# Defining Elimination of Genital Warts—A Modified Delphi Study

**DOI:** 10.3390/vaccines8020316

**Published:** 2020-06-18

**Authors:** Laila Khawar, Dorothy A. Machalek, David G. Regan, Basil Donovan, Skye McGregor, Rebecca J. Guy

**Affiliations:** 1The Kirby Institute, UNSW Sydney, Sydney, NSW 2052, Australia; dmachalek@kirby.unsw.edu.au (D.A.M.); Dregan@kirby.unsw.edu.au (D.G.R.); Bdonovan@kirby.unsw.edu.au (B.D.); Smcgregor@kirby.unsw.edu.au (S.M.); Rguy@kirby.unsw.edu.au (R.J.G.); 2Centre for Women’s Infectious Diseases, the Royal Women’s Hospital, Parkville, VIC 3052, Australia; 3School of Population and Global Health, University of Melbourne, Parkville, VIC 3053, Australia; 4Sydney Sexual Health Centre, Sydney Hospital, Sydney, NSW 2000, Australia

**Keywords:** human papillomavirus HPV, quadrivalent HPV vaccine, nonavalent HPV vaccine, 9-valent vaccine, genital warts, elimination, Australia’s HPV vaccination programme, elimination target, elimination as a public health problem, Delphi study

## Abstract

*Background:* Substantial declines in genital warts (GW) have been observed in countries with quadrivalent HPV vaccination programmes, with Australia showing the highest reductions due to early commencement and high vaccination coverage. There is a real potential to achieve GW elimination; however, no GW elimination definition exists. Taking Australia as a case study, we aimed to reach expert consensus on a proposed GW elimination definition using a modified Delphi process. *Method:* We used modelling and epidemiological data to estimate the expected number of new GW cases, from pre-vaccination (baseline) in 2006 to the year 2060 in Australian heterosexuals, men who have sex with men (MSM), and newly arrived international travellers and migrants. We used these data and the literature, to develop a questionnaire containing ten elimination-related items, each with 9-point Likert scales (1—strongly disagree; 9—strongly agree). The survey was completed by 18 experts who participated in a full day face-to-face modified Delphi study, in which individuals and then small groups discussed and scored each item. The process was repeated online for items where consensus (≥70% agreement) was not initially achieved. Median and coefficient of variation (COV) were used to describe the central tendency and variability of responses, respectively. *Findings:* There was a 95% participation rate in the face-to-face session, and 84% response rate in the final online round. The median item score ranged between 7.0 and 9.0 and the COV was ≤0.30 on all items. Consensus was reached that at ≥80% HPV vaccination coverage, GW will be eliminated as a public health problem in Australia by 2060. During this time period there will be a 95% reduction in population-level incidence compared with baseline, equivalent to <1 GW case per 10,000 population. The reductions will occur most rapidly in Australian heterosexuals, with 73%, 90% and 97% relative reductions by years 2021, 2030 and 2060, respectively. The proportion of new GW cases attributable to importation will increase from 3.6% in 2006 to ~49% in 2060. *Interpretation:* Our results indicate that the vaccination programme will minimise new GW cases in the Australian population, but importation of cases will continue. This is the first study to define GW elimination at a national level. The framework developed could be used to define GW elimination in other countries, with thresholds particularly valuable for vaccination programme impact evaluation. *Funding*: LK supported through an Australian Government Research Training Programme Scholarship; unconditional funding from Seqirus to support the Delphi Workshop.

## 1. Introduction

Human papillomavirus (HPV) is the most common sexually transmissible infection (STI) globally, and is associated with significant morbidity and mortality due to genital warts (GW), and anogenital and other cancers [1,2]. Three HPV vaccines are currently available, the bivalent (*b*HPV), quadrivalent (*q*HPV) and nonavalent (*n*HPV) vaccines [3]. All three vaccines provide protection against the high-risk HPV types 16/18 that are responsible for over 70% of cervical cancers [3]. The *q*HPV and *n*HPV vaccines also provide protection against low-risk HPV types 6/11 that cause more than 95% of GW [4], while the *n*HPV vaccine prevents infection for five additional cancer-causing types (31, 33, 45, 52 and 58) [3]. Over 80 countries have introduced an HPV vaccination programme, and the majority of these use the *q*HPV or *n*HPV vaccines. In 2018, the *q*HPV and *n*HPV vaccine together comprised ~77% of the total vaccine doses delivered globally [5].

The primary aim of HPV vaccination is to prevent cervical cancer, which is associated with high global mortality and morbidity [6]. In 2018, the World Health Organization (WHO) called for working toward the ambitious goal of eliminating cervical cancer as a public health problem [7]. Furthermore, in its Global Health Sector Strategy on STIs, the WHO highlights the importance of vaccination to achieve the elimination of GW and encourages countries to define national GW elimination targets [8]. Since the commencement of *q*HPV/*n*HPV vaccination, substantial reductions in GW have been observed in many countries [9], suggesting real potential for GW elimination. Notably, Australia, where a national programme commenced in 2007 for girls and 2013 for boys [10,11], has within a decade achieved 96% and 88% reductions in new GW diagnoses among young vaccine-eligible Australian-born women and heterosexual men attending sexual health clinics, respectively [12]. Australia is likely to be the first country to reach GW elimination; however, in terms of defining elimination, several points remain uncertain.

First, there is no standardised definition of GW elimination. The WHO broadly defines elimination of an infection or infectious disease under two categories: (i) interruption of transmission of an infection, defined as reduction to zero of the incidence of infection in a defined geographical area [13] and (ii) elimination as a public health problem, defined as achievement of measurable global targets set by the WHO [13]. It is unclear which of these elimination concepts is suitable for GW. Second, published modelling studies of GW elimination did not account for the potential impact of importation and spread of HPV 6/11 by people born overseas [14,15,16]. Third, there are uncertainties around the impact of GW cases caused by other non-vaccine HPV types on ongoing GW cases and the mechanisms by which this can be measured [17,18]. Finally, measuring elimination ideally requires a sustainable system to measure both incidence and the burden of disease including prevalence, and the associated morbidity and financial cost. Taking Australia as a case study, we conducted a modified Delphi study, firstly to reach expert consensus on how elimination should be defined and measured, and secondly, to develop GW control and elimination thresholds.

## 2. Method

### 2.1. The Delphi Method

The Delphi method has been widely used in public health to reach consensus among a group of experts in a specific subject area. In situations where available information is contradictory or insufficient, the Delphi method provides an effective tool to make decisions [19]. The method involves a structured multilevel-iterative process with a group of experts, where each iteration uses feedback from the preceding rounds. A conventional Delphi process entails four essential elements of iteration, anonymity, controlled feedback, and assessment of group judgement that provides an opportunity for the participants to revise their views [20]. This process continues until no further significant changes occur between rounds and a group consensus is achieved. The group Delphi study, otherwise known as an expert workshop, is a variation of the conventional Delphi method [20]. It preserves all the elements of a conventional Delphi except for anonymity, which is compromised to expedite the consensus-forming process.

### 2.2. Design and Participants

We designed a modified Delphi study (Figure 1) comprising a two-round face-to-face Delphi workshop with a group of national and international experts followed by an online round. Experts were selected using a purposive sampling technique to represent epidemiological, clinical, mathematical modelling, statistics, and health policy related expertise in HPV and other STIs.

### 2.3. Procedures

#### 2.3.1. Epidemiological Analyses

We first conducted a review of the literature (peer-reviewed and grey literature) and held extensive consultations with six Australian experts. Extending our previous work [14,15], we estimated the number of new GW cases among Australian heterosexuals and MSM, plus newly arrived international travellers, using epidemiological and modelling data at the baseline of 2006. The parameters included were population size, baseline GW incidence, sexual behaviour and mixing, and vaccination coverage in Australia and the home countries of international travellers [15,21,22,23,24,25,26,27]. Population size projection, population-specific GW incidence, and modelled relative reductions in GW due to vaccination were calculated for the years 2021, 2030 and 2060, based on the timepoints presented in our previous research [14]. This led to the development of a proposed elimination threshold for GW for the year 2060. In addition, we also proposed two disease-control related thresholds for the years 2021 and 2030 to monitor the trajectory towards GW elimination (Table 1). The background calculations for the proposed thresholds are shown in Appendix A.

#### 2.3.2. Questionnaire Development

The epidemiological analyses resulted in the development of a questionnaire containing preliminary GW elimination definitions and thresholds. The questionnaire was tested in a small pilot Delphi with eight early- to mid-career level researchers to ensure the clarity and acceptability of the items. An eight-item questionnaire (see Appendix A for details) was then administered in the Delphi workshop. Subsequently, the questionnaire was revised based on experts’ feedback at the end of each round of the study and more items were added accordingly.

#### 2.3.3. Delphi Workshop

Prior to the workshop, experts were sent a document explaining the aims of the study and the methods used for calculation of the proposed thresholds. The workshop began with a plenary session where the questionnaire and the process of reaching consensus was introduced. In round 1, experts were asked to score items individually in terms of their agreement on a 9-point Likert scale, with 1 being strongly disagree, and 9 being strongly agree. Experts were also asked to provide qualitative feedback on each item. All responses were treated anonymously, and therefore even when the participants were known to one another, their opinions remained unidentifiable to the group.

In round 2, experts were divided into four small groups of 4–5 people and asked to score their agreement or disagreement as a group to help consolidate opinions in a short time period. Each group was also asked to provide qualitative feedback to justify their viewpoint. Results were presented during plenary sessions after each round. Convergence and deviation of opinions were brought forward in the plenary sessions and groups were asked to justify any conflicting opinions for discussion with the whole group.

#### 2.3.4. On-Line Round

Items that did not reach consensus at the workshop were revised based on experts’ feedback. A detailed document explaining the revised items was then distributed via email. A weblink to a survey was provided and experts were asked to score the revised items.

### 2.4. Data Analysis and Level of Consensus

Median and mean were used to describe the central tendency, and coefficient of variation (COV) and interquartile ranges were used to describe the variability of expert responses on each item. Items were excluded from the subsequent round if ≥70% of participants scored the item ≤4 in the preceding round with a COV of ≤0.5. Items met consensus for inclusion if ≥70% of participants scored the item ≥7 in the preceding round with a COV of ≤0.5. If items failed to meet the inclusion/exclusion criteria in the preceding round, they were modified as per the feedback for the successive round. A response rate of 80% for each round of the study was considered sufficient.

### 2.5. Ethics Approval

Ethical approval was obtained from the UNSW Human Research Ethics Advisory Panel (reference number HC16530). All experts were provided with a participation information sheet. Consent was obtained prior to their participation in the workshop.

### 2.6. Role of Funding Source

The sponsor had no role in study design, data collection, analysis, interpretation or writing of the report. The corresponding author has full access to all the data and had full responsibility to submit for publication.

## 3. Results

### 3.1. Characteristics of the Experts

We invited 21 experts to participate in the study, of whom 19 agreed. Of these, 18 participated in the workshop (95%), and 16 participated in the online round (84%). The characteristics of the experts are shown in Table 2. The Delphi participants were mostly female, senior academics and from a range of disciplines (Table 2).

### 3.2. Results of Delphi Rounds 1–3

#### 3.2.1. Conceptual Definitions (Items 1 and 2)

At the end of the workshop, two of four groups of experts scored the concept of interruption of endemic transmission of HPV 6/11 ≤ 4 (median:4.5; COV: 0.65) (Table 3). Two conflicting themes emerged, evident by a large COV. According to one group, this was an ambitious goal which would be very difficult to measure. They emphasized that measuring elimination using this definition is predicated on the ability to differentiate between cases attributable to local strains circulating in the community (endemic) versus imported cases. The opposing group argued that interruption of endemic transmission of HPV 6/11 was an aspirational goal necessary to promote vaccine coverage. After the workshop, we reviewed the relevant literature, which indicated that discrete geographical variations of HPV 6/11 subtypes do not exist, therefore distinguishing between endemic and imported cases through genotyping is not possible (Appendix A). A revised item was presented in the online round, with consensus reached that interruption of endemic HPV 6/11 transmission is not a feasible elimination definition (median 7.0; COV, 0.20) (Table 4).

Similarly, for the concept of elimination as a public health problem, two of the four groups of experts agreed on this definition by scoring it ≥ 7 (median: 6.5; COV: 0.31) (Table 3). Some participants argued that this is a pragmatic definition, while others maintained that this is not as aspirational as interruption of transmission of infection. Nonetheless, there was general agreement on the need to clarify what constituted a public health problem, and whether GW fulfilled the criteria.

A review of literature was performed post workshop through which it was determined that the reported prevalence of GW in the pre-vaccination era satisfies the essential criteria for classification of a condition as a public health problem (Appendix A), and the relevant details were communicated to experts. A revised item was presented in the online round where consensus of 81% was reached for inclusion (median: 8.0, COV: 0.20) (Table 4).

#### 3.2.2. Operational Definitions/Thresholds (Items 3–6 and 10)

Views were divided on the potential impact of non-vaccine HPV types on the control threshold (≤4 cases per 10,000 population by year 2030, equivalent to 82% relative reduction from the baseline of 2006) and the elimination threshold (≤1 case per 10,000 population by year 2060, equivalent to 95% relative reduction from the baseline of 2006) at the end of the workshop (Table 3). Some experts considered there to be a need to undertake HPV type surveillance of lesions to answer this question, while others argued that the small proportion of GW attributable to non-vaccine HPV types was unlikely to have an impact on the estimated thresholds. Following the workshop, new evidence was published showing significant cross-protection against non-vaccine HPV types, including a majority of those that cause GW [28] (Appendix A). As the intention of the Delphi process was to address questions where there is an absence of evidence, this new evidence made items 3 and 5 redundant for the online round, and these items were therefore removed from the online round.

Similarly, at the end of the workshop, views were divided regarding the potential impact of ongoing transmission of GW due to importation on the estimated control and elimination thresholds (Table 3). Some experts believed that chains of transmission due to importation could have an impact, and that modelling is required to accurately calculate the HPV 6/11 effective reproductive number (R*_eff_*). R*_eff_* denotes the average number of secondary cases per infectious case in a population made up of both susceptible and non-susceptible hosts. Other experts argued that the effects could potentially be less than expected due to growing levels of HPV vaccination globally. Nonetheless, experts suggested that perhaps thresholds needed to be adjusted to account for this. The experts also suggested defining an additional control milestone for the year 2021.

Based on the experts’ feedback, we conservatively adjusted our proposed control milestone (≤7 cases per 10,000 population, equivalent to 67% relative reduction from the baseline of 2006) and control threshold for the years 2021 and 2030, respectively, to account for the effects of chains of transmission due to importation. The relative reductions in new GW diagnoses were changed from 67% to 60% for 2021 and from 82% to 80% for 2030 (Table 5). The elimination threshold for 2060 (≤1 case per 10,000 population, equivalent to 95% relative reduction from the baseline of 2006), however, was not adjusted. Revised items were presented in the online round. Items 4 and 6 met consensus at 81% (median 7.5, COV: 0.27) and 75% (median 7.0, COV: 0.24), respectively (Table 4). This signified a convergence of opinion for the 2030 control threshold and 2060 elimination threshold. An additional item on a control milestone for the year 2021 was added as per the feedback and met consensus for inclusion (Table 4).

#### 3.2.3. Intervention-Coverage (Item 7)

In round 2, experts suggested rewording the item relating to vaccination coverage to reflect completion of HPV vaccination course rather than two dose completion, as two dose coverage (0–2 months apart) for the *q*HPV vaccine schedule may be different from two dose coverage (0–6 months apart) for the *n*HPV vaccine, which replaced the *q*HPV vaccine in the national HPV vaccination programme in Australia in 2018. At the end of round 2, the reworded item met consensus for inclusion at 75% (median 8.0, COV: 0.16) (Table 3).

#### 3.2.4. Elimination Measurement (Items 8 and 9)

At the end of round 1, most experts disagreed with using sexual health clinics as the main data source for measuring GW elimination and suggested that both general practice and sexual health clinic data should be used, the two out-patient healthcare settings in Australia where GW episodes are seen. The revised item, including both data sources, met consensus for inclusion at 75% (median 9.0; COV 0.18) (Table 3). Experts also suggested an additional item on genotyping of GW. Therefore, a new item was added in round 2, indicating that measurement should include genotyping of GW to better understand the aetiology of remaining warts, which reached unanimous consensus (median 8.0; COV 0.14) (Table 3).

#### 3.2.5. Revised Background Calculations

After adjusting for ongoing transmission due to importation, our background calculations to inform these thresholds showed that the fastest reductions will occur in Australian resident heterosexuals, with ~73%, ~90% and ~97% relative reduction by years 2021, 2030 and 2060, respectively (Figure 2). Reductions in MSM will be slower initially, with a slight increase by 2021, followed by a ~37% and ~99% relative reduction by the years 2030 and 2060, respectively. Our analyses also demonstrated that the proportion of all new GW cases in Australia attributable to importation will increase from 36% in 2006 to ~49% in 2060 (Figure 3). The relative reduction in new GW diagnoses in international travellers will vary depending on the vaccination coverage in their home countries. As a majority (78%) of backpackers are from countries with a *q*HPV/*n*HPV vaccination programme (Appendix A), this group will observe the most substantial drop in GW incidence (Figure 3). On the other hand, the reduction in GW in international students is predicted to be slower and less substantial, as a majority (72%) are from countries with no current HPV vaccination programme or with *b*HPV vaccination programmes that do not protect against GW. It was estimated that nearly half of the cases in the international travellers will be attributable to international students by 2060.

## 4. Discussion

To the best of our knowledge, this is the first study to define GW elimination at a national level (using Australia as a case study). In the past, elimination targets of other infectious diseases have been presumably developed in consultation with experts, but often very little detail of the process is available [8]; this is the first documented instance of a Delphi study being used to define elimination of an infectious disease. Consensus was reached that with an ongoing HPV vaccination programme achieving ≥80% coverage in the target population, GW will be eliminated as a public health problem in Australia by 2060 at ≤1 new GW case per 10,000 population, equivalent to a 95% reduction in GW incidence as compared to the baseline of 2006. This impact will be observed at a population level including all Australian residents and recently arrived international travellers. Consensus was also met on the need for surveillance of GW diagnoses at all primary health services that are representative of the Australian population and that this should include genotyping. Our study suggests that while major declines have already been observed in vaccine-eligible Australian-born people, population-level elimination will take several decades, partly due to GW importation from countries without a *q*HPV or *n*HPV vaccine programme.

Cervical cancer is a major HPV-related global public health problem and the aspirational goal of elimination defined by the WHO provides impetus for countries to achieve high HPV vaccination coverage using one of three available HPV vaccines. The primary purpose of HPV vaccination is to prevent cervical cancer, and while all three vaccines are highly effective in providing protection against infections with cancer causing HPV types 16/18, two HPV vaccines, *q*HPV and *n*HPV, also provide protection against GW caused by low-risk HPV types 6/11. The choice of vaccine for a national vaccination programme generally depends on affordability, but in some countries, such as the United Kingdom, the decision was influenced by our previous research [29] on the reductions in genitals warts achieved by using the *q*HPV vaccine [30]. In the event a country chooses to adopt the *q*HPV or the *n*HPV vaccine for their national programme, defining GW elimination thresholds will provide useful additional information to monitor the programmes’ performance. The framework developed through our study for Australia could be replicated for other countries, if comprehensive epidemiological and mathematical modelling information is available in the local context.

Our HPV vaccination course completion threshold of ≥80% coverage necessary to reach population-level elimination in Australia, is consistent with the WHO target of 90% for female-only vaccination [7] and the 80% target specified in the Fourth Australian National STI Strategy target for adolescent HPV vaccination [31]. This vaccination threshold serves as a benchmark to reach GW elimination and does not negate the vaccination threshold set for elimination of cervical cancer, which requires high and wide-spread coverage of both HPV vaccination and cervical screening [7].

Our results suggest that interruption of endemic transmission of HPV 6/11 is not a suitable elimination definition for GW. For other infections for which elimination thresholds have been set by the WHO for interruption of endemic transmission (i.e., malaria, measles, and congenital rubella syndrome), case-based surveillance is carried out by conducting genotyping in every outbreak to differentiate endemic versus imported or import-related cases [32,33]. Such discrete geographical variations of HPV 6/11 subtypes do not exist [34,35,36], and therefore even if interruption of endemic HPV 6/11 is achieved in the future, distinguishing between endemic and imported cases through genotyping may not be possible.

Elimination as a public health problem is a more appropriate approach for defining GW elimination. The burden of GW in Australia prior to the introduction of *q*HPV vaccination satisfied the essential criteria for classification of a condition as a public health problem [37,38]—namely, high burden of disease [21], loss of quality of life [39], high financial cost [21], and the feasibility to act at a community level [29]. While our results indicate that GW will be eliminated as a public health problem by year 2060 at a population level with the fastest reduction in Australian resident heterosexuals, it will be important to monitor progress in certain sub-populations that have historically been at higher risk of GW, and have not yet demonstrated significant declines in GW, such as immunocompromised individuals [40], transgender people [41], and migrant sex workers [42]. This will ensure that long-term benefits extend to all individuals in an equitable manner and that targeted interventions can be implemented where necessary.

A key finding from our study is that absolute elimination of GW in Australia is unlikely. Firstly, importation of GW caused by HPV6/11 will continue with over half the remaining cases occurring in international travellers in 2060. Of these, nearly half the cases will be attributed to international students. Currently, a majority (72%) of Australian international students are from countries that either do not have an HPV vaccination programme or only have a *b*HPV vaccination programme, with China and India (both currently with no HPV vaccination programme) comprising 24% and 13% of international students, respectively [43]. Therefore, there is a need to monitor GW prevalence over time in this group of travellers to evaluate the extent to which they might undermine the impact of the Australian HPV vaccination programme.

Secondly, a small proportion of GW are caused by HPV types for which current HPV vaccines provide no direct protection. As noted, the control and elimination thresholds defined in this study were not adjusted for the contribution of GW caused by non-vaccine HPV types. The underlying assumption was that their contribution would likely be minimal or insignificant based on new evidence from a study that compared urine samples of vaccinated and unvaccinated cohorts of young girls and suggested that *q*HPV vaccination offers significant cross-protection against non-vaccine HPV types, including the majority of those that cause GW [28]. In the light of this evidence and in the absence of evidence of type replacement [44], we concluded that GW attributable to non-vaccine HPV types may also decline. Nevertheless, there is a possibility that we have underestimated the impact of non-vaccine HPV types as HPV prevalence may be lower in urine samples than cervical or vaginal samples [45,46,47]. To truly understand the pathogenesis and causality of non-vaccine HPV types in GW in the post-vaccination era, a prospective study with adequate power is required to detect specific HPV genotypes in GW.

We conservatively adjusted the control milestone and threshold to account for the impact of ongoing chains of transmission in Australia due to importation of HPV types that cause GW. The underlying assumption was that population-level herd protection would not be reached by years 2021 and 2030 and therefore transmission due to importation will continue. One way of addressing this question is to calculate R*_eff_* for HPV 6/11 and then use this to estimate the probability that a chain of transmission due to importation will occur. This is outside the scope of this study; however, it is the subject of ongoing work by the authors. We did not adjust the 2060 elimination threshold of 95% relative reduction in GW incidence as the HPV vaccination coverage is likely to increase over time, both in Australia and globally, and the importance of importation is likely to be diminished accordingly. This is particularly important in terms of the anticipated increase in vaccination coverage in low- and middle-income countries where HPV vaccination might become more affordable due to: (a) technology transfers of existing vaccines to manufacturers in middle-income countries [48,49]; and (b) moving to single-dose vaccination, which is currently being evaluated in ongoing studies [50].

Our study has some limitations. First, we compromised the anonymity of experts in the workshop as this was necessary to expedite the consensus building process. In doing so, we may have introduced a degree of ‘bandwagon effect’, a phenomenon whereby those with strong opinions may sway the opinions of others [51]. To counter this effect, we provided an opportunity for anonymous individual scoring in the third online round so that the experts could reflect on and score the revised items without peer pressure. Second, where information was not available for baseline GW incidence and relative reduction of GW incidence, we used baseline GW prevalence and relative reduction in incidence or prevalence of HPV 6/11 as proxies (Appendix A). In doing so, we might have overestimated the number of new GW cases and the estimated relative reductions in some of the sub-populations.

Our study also has several strengths and fulfilled important factors that determine the validity of the Delphi technique. These include: (a) selection of a heterogenous panel, with expertise across various areas of the health sector [52]; (b) achievement of a high response rate [53]; (c) retention of those experts that held outlying opinions (important due to their potential for challenging conventional thinking) [52]; and (d) careful attention to content analysis and interpretation of the survey data [19,54] that took into account the qualitative feedback collected during each round and the two plenary sessions. The last point is particularly important, as our study took into consideration the richness and depth found in ‘live’ group discussion, which is often missing in conventional Delphi studies.

In conclusion, we defined thresholds for elimination of GW at a national level, using Australia as the case study, the first country in the world to implement a nationally funded *q*HPV vaccination programme [55]. On the assumption that Australia maintains its high HPV vaccination coverage, GW elimination defined through this study will serve as a valuable tool for programme impact evaluation. However, questions remain around the potential impact of non-vaccine HPV types and the ongoing chains of transmission due to importation. This means that the control and elimination thresholds defined by this study may need to be adjusted in the future subject to research that may provide new information. As more countries roll out HPV vaccination programmes, the framework developed through this study could be used to help guide the development of GW elimination thresholds in other settings.

## Figures and Tables

**Figure 1 vaccines-08-00316-f001:**
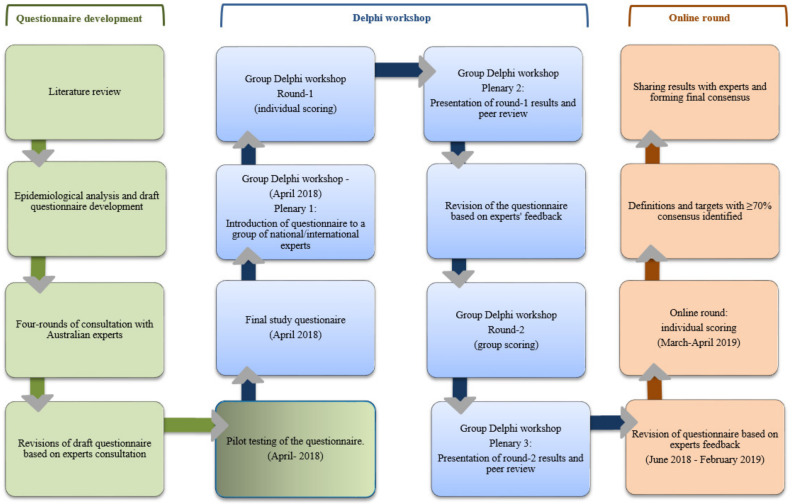
Flowchart of the process for development of consensus.

**Figure 2 vaccines-08-00316-f002:**
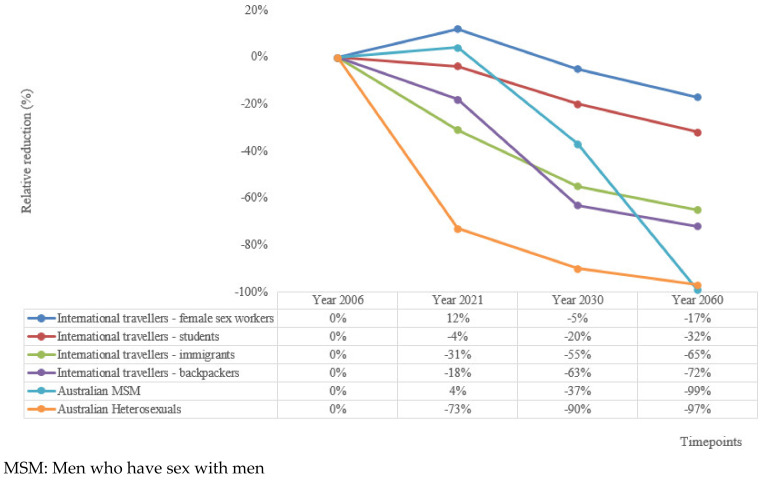
Revised estimates of relative reduction in new cases of genital warts in Australia, in all populations considered, after adjusting for the ongoing transmission due to importation of genital warts, by time period.

**Figure 3 vaccines-08-00316-f003:**
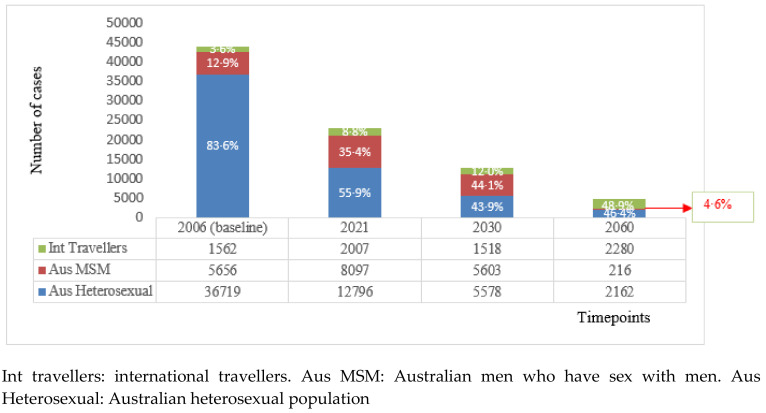
Revised number and proportion of estimated new genital warts cases in Australia for timepoints 2006, 2021, 2030 and 2060, after adjusting for the ongoing transmission due to importation of genital warts, by population type.

**Table 1 vaccines-08-00316-t001:** Estimates presented at the Delphi workshop for number and relative reduction in new genital warts cases in Australia (consolidated cases including Australian residents and international travellers).

	Australian	New Genital Warts Cases—*n* (95% CI)				Rate Per 10,000 Persons	Relative Reduction (%)
	Population					(95% CI)	in New Genital Warts
		International	Australian Heterosexual	Australian MSM	Consolidated Cases		from Baseline (95% CI)
		Travellers	Population	Population			
Baseline (2006)	20,091,504	1562 (1428–1707)	36,719 (31,579–42,181)	5656 (5512–5803)	43,937 (38,519–49,691)	21.9 (19.2–24.7)	..
2021	26,110,176	1634 (1492–1788)	10,419 (9849–13,155)	6593 (6437–6752)	18,647 (17,777–21,695)	7.1 (6.8–8.3)	67.3 (66.8–67.9)
2030	29,748,172	1364 (1232–1510)	5012 (4350–5811)	5034 (4897–5174)	11,410 (10,479–12,494)	3.8 (3.5–4.2)	82.4 (82.1–82.8)
2060	40,703,739	2280 (2110–2462)	2162 (1876–2506)	216 (188–247)	4658 (4174–5215)	1.1 (1.0–1.3)	94.7 (94.6–94.9)

Estimates calculated without accounting for new genital warts cases attributable to non-vaccine type HPV infection and the chains of transmission in Australia due to importation of genital warts. (..) = not applicable.

**Table 2 vaccines-08-00316-t002:** Characteristics of experts who completed the Delphi workshop and the online round.

	Delphi Workshop (*n* = 18)	Online Round (*n* = 16)
Gender		
Female	72%	69%
Male	28%	31%
Age group		
≤35 years	11%	13%
35–44 years	22%	25%
45–54 years	50%	50%
≥55 years	17%	13%
Education		
Masters	6%	6%
Medical degree	11%	6%
PhD	83%	88%
Professional level		
Professional (non-academic)	11%	6%
Early-/mid-career academic	22%	25%
Senior academic	67%	69%
Expertise		
Health policy	6%	6%
Statistics	6%	6%
Mathematical modelling	22%	25%
Vaccination/immunology	22%	13%
Epidemiology	22%	25%
Clinical and epidemiology	17%	19%
Other	6%	6%

**Table 3 vaccines-08-00316-t003:** Questionnaire and results of round 2 of the Delphi face-to-face workshop.

Revised Items for Round 2 of the Delphi Workshop	Median	Mean	% Agreement	% Disagreement	Outcome
	(IQR)	(COV)	(Scores 7,8 & 9)	(Scores 1,2,3 & 4)	
**Section-1: Conceptual definitions**					
**Item-1: Elimination of (endemic) transmission:** Interruption of endemic genital warts transmission caused by HPV 6 and 11, and limited transmission from imported cases.	4.5	4.3	25%	50%	To revise item as per
	(2.5–6.3)	(0.65)			experts’ feedback
**Item-2: Elimination as a public health problem:** Transmission of genital warts continues to occur (even in absence of importation) but is reduced to a level that it does not constitute a public health problem.	6.5	6.7	50%	25%	To revise item as per
	(5.5–7.5)	(0.31)			experts’ feedback
**Section-2: Operational thresholds**					
*Short-term control threshold: ≤**4 cases per 10,000 population by year 2030;* Equivalent to: *A reduction in annual genital warts incidence by 82% by year*	..	..	..	..	..
*2030*					
*Long-term elimination threshold: ≤1 case per 10,000 population by year 2060;* Equivalent to: *A reduction*	..	..	..	..	..
*in annual genital warts incidence by 95% by year 2060*					
**Item 3:** The proportion of genital warts caused by non-vaccine HPV types will have an impact on the control threshold in year 2030	2.5	3.5	25%	75%	To revise item as per
	(1.7–4.2)	(0.89)			experts’ feedback
**Item 4:** Ongoing transmission of genital warts in Australia due to importation would have an impact on the control threshold in year 2030	3.5	3.8	25%	50%	To revise item as per
	(1.7–5.5)	(0.70)			experts’ feedback
**Item 5:** The proportion of genital warts caused by non-vaccine HPV types will have an impact on the elimination threshold in year 2060	6.0	5.8	50%	25%	To revise item as per
	(4.2–7.5)	(0.52)			experts’ feedback
**Item 6:** Ongoing transmission of genital warts in Australia due to importation would have an impact on the elimination threshold in year 2060	3.5	4.0	25%	50%	To revise item as per
	(1.7–5.7)	(0.79)			experts’ feedback
**Section-3 Intervention coverage/Process threshold**					
**Item 7:** Completion of HPV vaccination course is equal to or greater than 80% coverage in the target population	8.0	7.8	75%		Consensus met–
	(7.5–8.2)	(0.16)			accept
**Section-4: Measuring elimination**					
**Item 8:** There needs to be measurement of genital warts from both general practice and sexual health clinics	9.0	8.3	75%	..	Consensus met–
	(8.2–9.0)	(0.18)			accept
**Item 9 (new item):** Measurement should also include genotyping of genital warts	8.0	8.0	100%	..	Consensus met–
	(7.0–9.0)	(0.14)		..	accept

(..) = not applicable.

**Table 4 vaccines-08-00316-t004:** Questionnaire and results of the third online round.

Revised Items for the Online Round	Median	Mean	% Agreement	% Disagreement	Outcome
	(IQR)	(COV)	(Scores 7,8 & 9)	(Scores 1,2,3 & 4)	
**Conceptual definitions**					
**Item 1: Elimination of (endemic) transmission**: Defining genital warts elimination as interruption of endemic HPV 6 & 11 transmission is not feasible due to a lack of geographic variation in HPV 6&11 subtype distribution	7.0	6.9	81%	6%	Consensus met–
	(7.0–8.0)	(0.20)			agreed to reject
**Item 2: Elimination as a public health problem**: To reduce the burden of genital warts to a level where it no longer constitutes a public health problem	8.0	7.4	81%	6%	Consensus met–
	(7.0–9.0)	(0.20)			accept
**Operational thresholds^**					
**Item 10 (new item): Control milestone for the year 2021**—A 60% relative reduction in new genital warts diagnoses by year 2021 at a population level, as compared to the baseline of 2006; Equivalent to: Reduction of new cases of genital warts to ≤9 cases per 10,000 population by year 2021	7.5	6.8	75%	13%	Consensus met–
	(6.8–8.0)	(0.30)			accept
**Item 4*: Control threshold for the year 2030**—An 80% relative reduction since 2006 in new genital warts diagnoses by year 2030 at a population level, as compared to the baseline of 2006; Equivalent to: Reduction of new cases of genital warts to ≤4 cases per 10,000 population by year 2030	7.5	6.9	81%	13%	Consensus met–
	(7.0–8.0)	(0.27)			accept
**Item 6*: Elimination threshold for the year 2060**—A 95% relative reduction since 2006 in new genital warts diagnoses by year 2060 at a population level, as compared to the baseline of 2006 Equivalent to: Reduction of new cases of genital warts to ≤1 case per 10,000 population by year 2030	7.0	7.0	75%	6%	Consensus met–
	(6.8–8.0)	(0.24)			accept

^ Items 3 and 5 were removed from the online round based on emerging evidence—see Appendix A. * After adjusting for ongoing short chain of transmission of genital warts in Australia due to importation.

**Table 5 vaccines-08-00316-t005:** Revised estimates for number and relative reduction in new genital warts consults in Australia, after adjusting for the ongoing transmission due to importation of genital warts (consolidated cases including Australian residents and international travellers).

	Australian	New GW Cases—*n* (95% CI)				Rate Per 10,000 Persons	Relative Reduction (%)
	Population					(95% CI)	in New Genital Warts from
		International	Australian Heterosexual	Australian MSM	Consolidated		Baseline (95% CI)
		Travellers	Population	Population			
Baseline (2006)	20,091,504	1,562 (1,428–1,707)	36,719 (31,579–42,181)	5656 (5512–5803)	43,937 (38,519–49,691)	21.9 (19.2–24.7)	..
2021	26,110,176	2007 (1849–2177)	12,796 (11,255–15,034)	8097 (7924–8072)	22,900 (21,029–25,483)	8.8 (8.1–9.8)	59.9 (59.2–60.5)
2030	29,748,172	1518 (1377–1670)	5578 (5037–6728)	5603 (5459–5750)	12,700 (11,873–14,149)	4.3 (4.0–4.8)	80.4 (80.1–80.9)
2060	40,703,739	2280 (2110–2462)	2162 (1876–2506)	216 (188–247)	4658 (4174–5215)	1.1 (1.0–1.3)	94.8 (94.6–94.9)

(..) = not applicable.

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
