# Peer review of "Defining Elimination of Genital Warts—A Modified Delphi Study"

_vaccines, 2020, doi:10.3390/vaccines8020316_

Round 1
Reviewer 1 Report
The authors present an interesting study which describes the use of an expert study (Delphi study) to define the elimination of genital warts (GW) as a public health issue as a consequence of human papillomavirus vaccination (HPV) programs. The approach has used Australia as the test country as it was the first to adopt HPV vaccination. The authors used a series of expert workshops to reach a consensus about how this might be achieved and what it might "look" like. It was agreed that an HPV vaccination coverage of greater than 80% was required, as this would reduce the incidence of GW by 95% in 2060 compared to 2006 rates (prevaccination). This would equate to less than 1 new GW case per 10,000 population. It was also agreed that as a consequence of various factors, including importation of cases, from overseas it unlikely that GW would be eradicated. Overall the study is well written and the conclusions drawn are supported by the data provided.
I would suggest that authors revise the abstract to include their final conclusion, i.e. "less than 1 new GW case per 10,000 population by 2060" - this one of the key findings of the study.
I would also suggest the authors consider modifying the title of the manuscript. The current title suggests the manuscript will define the "elimination of genital warts" - however, this is somewhat inconsistent with the text. The manuscript moves towards a new target where GWs are not eradicated but rather move to the modified status of no being a public health concern.
Table 1
Row 5, Column 4, and Column 7 - the 95% confidence intervals are incomplete.
Author Response
We would like to thank the reviewers for their valuable feedback. We have addressed the points highlighted by the reviewers. Please see our comments below:
REVIEWER 1:
Point 1: “I would suggest that authors revise the abstract to include their final conclusion, i.e. "less than 1 new GW case per 10,000 population by 2060" - this one of the key findings of the study”.
Author’s reply: Thank you for this suggestion. We have added this in the abstract, line 36.
Point 2: “would also suggest the authors consider modifying the title of the manuscript. The current title suggests the manuscript will define the "elimination of genital warts" - however, this is somewhat inconsistent with the text. The manuscript moves towards a new target where GWs are not eradicated but rather move to the modified status of not being a public health concern”.
Author’s reply: We appreciate this feedback but also note why we chose to use the term “elimination” in the title. The World Health Organiszation broadly defines elimination under two categories 1. Interruption of transmission of infection, and 2. Elimination as a public health problem. The former demands local eradication or bringing down the incidence to zero in a defined geographical area. The latter doesn’t require the incidence to come down to zero, but to an acceptable pre-defined level. We have defined elimination of genital warts using the latter definition and would therefore contend that this is in alignment with our title.
Point 3: “Table 1 - Row 5, Column 4, and Column 7 - the 95% confidence intervals are incomplete”
Author’s reply: It was a formatting error with the row width hindering the view. We have adjusted the row width and the CIs are now visible.
Reviewer 2 Report
This study sought expert opinions to define outcomes for elimination of genital warts caused by HPV and how these outcomes should be measured using a modified Delphi method using Australian data and population estimates. The study was thoughtfully designed and conducted and the results and discussion are well written. I have no concerns.
Author Response
We would like to thank the reviewer for their valuable feedback.
Reviewer 3 Report
General comments
This paper shows a GW elimination definition using a modified Delphi method. Since a standardized definition of GW elimination does not exist, authors tried to define and measure it. Ahead of the world, Australia has promoted a national programme and succeeded in reducing new GW diagnoses among young vaccine-eligible people. This study is based on such national approach, and the results obtained are considered to be convincing.
I would like to make some confirmations and recommendations regarding the framework of method adopted in this study.
1. About group Delphi
Delphi method has many variations, and the group Delphi is one of them. Is there any past example where the group Delphi has helped to formulate public health policy compared to standard Delphi method? Citing past example may increase the relevance of this study. In addition, it is better to describe this point in the interpretation part of the abstract.
2. Year of elimination (METHOD, Procedure)
Why did you set the endpoint for elimination in 2060? And why didn’t you set the milestone during 2030-2060? Authors should describe the reason in METHOD.
3. Experts (Table 2)
1) Is there any mention of the low proportion of non-academia compared to academia?
2) Are the ‘health policy’ experts policy researchers or administrative officers? These experts need to be distinguished.
Author Response
We would like to thank the reviewers for their valuable feedback. We have addressed the points highlighted by the reviewers. Please see our comments below:
REVIEWER 3:
Point 1: “About group Delphi - Delphi method has many variations, and the group Delphi is one of them. Is there any past example where the group Delphi has helped to formulate public health policy compared to standard Delphi method? Citing past example may increase the relevance of this study. In addition, it is better to describe this point in the interpretation part of the abstract”.
Author’s reply: Thank you for your feedback. Although, there are conventional Delphi studies published in the arena of public health, to the best of our knowledge, there is no published group Delphi study within the area of public health policy.
Point 2: “Year of elimination (METHOD, Procedure) - Why did you set the endpoint for elimination in 2060? And why didn’t you set the milestone during 2030-2060? Authors should describe the reason in METHOD.”
Author’s reply: We set the elimination timepoint in 2060, based on our previous mathematical modelling study that presented two milestones in years 2021, 2030 and a near-elimination timepoint at 2060. We have added a line in the Procedures (line 124) to clarify this point.
Milestones are usually setup to monitor the progress towards an assigned goal. In our case, our first assigned goal is in 2030, meeting it would signify the control of genital warts. To monitor our progress towards this target, we have a 2021 milestone. We did not set up a milestone after the 2030 control target due to the prolonged time period between 2030 and 2060, which, could potentially be reduced, based on our surveillance of genital warts in Australia in the future, requiring revised benchmarks.
Point 3: “Experts (Table 2) - 1) Is there any mention of the low proportion of non-academia compared to academia? 2) Are the ‘health policy’ experts policy researchers or administrative officers? These experts need to be distinguished.”
Author’s reply: Regarding the first point highlighted by the reviewer, we used purposive sampling technique to select experts in the field of epidemiology, mathematical modelling, statistics, clinical and health policy related expertise in HPV and sexually transmittable infections that yielded this result. Apparently, most experts in the field are also academics, hence the higher proportion as compared to professionals. For the second point, all experts with expertise in health policy are experts/researchers in health policy and not administrative officers.